# Word synonym relationships for text analysis: A graph-based approach

**Hend Alrasheed**  *

Department of Information Technology, College of Computer and Information Sciences, King Saud University, Riyadh, Saudi Arabia

* halrasheed@ksu.edu.sa

## Abstract

Keyword extraction refers to the process of detecting the most relevant terms and expressions in a given text in a timely manner. In the information explosion era, keyword extraction has attracted increasing attention. The importance of keyword extraction in text summarization, text comparisons, and document categorization has led to an emphasis on graph-based keyword extraction techniques because they can capture more structural information compared to other classic text analysis methods. In this paper, we propose a simple unsupervised text mining approach that aims to extract a set of keywords from a given text and analyze its topic diversity using graph analysis tools. Initially, the text is represented as a directed graph using synonym relationships. Then, community detection and other measures are used to identify keywords in the text. The set of extracted keywords is used to assess topic diversity within the text and analyze its sentiment. The proposed approach relies on grouping semantically similar candidate words. This approach ensures that the set of extracted keywords is comprehensive. Differing from other graph-based keyword extraction approaches, the proposed method does not require user parameters during graph construction and word scoring. The proposed approach achieved significant results compared to other keyword extraction techniques.

**Data Availability Statement:** All text datasets are available at https://github.com/halrashe/Topic-Diversity.

**Funding:** The author(s) received no specific funding for this work.

## Introduction

Currently, social media outlets produce extremely large amounts of data. Text analysis provides an effective way to process and utilize the most relevant data. Such analysis supports various applications in different domains, such as marketing, content filtering, and search. Manual processing of the huge number of documents available online is tedious, time-consuming, and error-prone. Text mining refers to the automatic extraction of information and the identification of valuable and previously unknown hidden patterns from unstructured textual data [1]. Text mining algorithms make it possible to process huge amounts of unstructured textual data efficiently and effectively.

Many text mining techniques, such as text summarization, text comparisons, document categorization, and document similarity measurements, depend on the extraction of a representative set of keywords from the given text [2, 3]. Keywords can be defined as a set of one or more

**Competing interests:** The authors have declared that no competing interests exist.

words that provides a compact representation of the content of a text document [4]. Automatic keyword extraction refers to the process of detecting the most relevant terms and expressions from a given text in a timely manner. Keyword extraction approaches can be categorized as statistical [5–7], machine learning [8, 9], linguistic [10], and graph-based approaches [3, 11–18]. Due to their simplicity, statistical approaches, such as term frequency, do not always produce good results [12]. Machine learning approaches require data training and can be biased toward the domain on which they were trained [19]. Graph-based keyword extraction techniques can capture more structural information about the text compared to other classic text analysis methods [20].

The underlying principle in graph-based keyword extraction is measuring and identifying the most important vertices (words) based on information obtained from the structure of the constructed text graph. Such vertices can be obtained using node centrality measures, such as degree centrality, closeness centrality, PageRank [12, 13, 15, 18] and $k$-degeneracy [18, 21]. However, keyword extraction approaches vary according to their text graph construction techniques, which directly impacts the ranking of the candidate keywords. Most of the proposed graph-based keyword extraction approaches depend on word co-occurrences; therefore, they do not necessarily generate a set of keywords that covers the main topics discussed in the text [22]. Moreover, most existing graph-based keyword extraction approaches require user parameters [12, 18, 21].

In this paper, we propose a simple unsupervised text mining approach that aims to extract a set of key concepts from a given text and then analyze its topic diversity using graph analysis tools. The proposed approach relies on grouping semantically similar candidate words; as a result, the extracted set of keywords is ensured to be comprehensive. Moreover, differing from other existing graph-based keyword extraction approaches [12, 18, 21], the proposed method does not require user parameters during graph construction and word scoring.

The text is first represented as a directed synonym graph. A word $v$ is a direct synonym of another word $u$ if it has a similar meaning. Word $v$ is an indirect synonym of word $u$ if a word $w$ that is a synonym of $v$ is also a synonym of word $u$. For example, the word "publication" is a direct synonym of the word "book" while the word "paper" is an indirect synonym of the word "book" (Fig 1). Direct synonym relationships between word pairs represent stronger relationships compared to indirect ones. Once the text graph is constructed, community detection and other measures are used to identify keywords. The set of most central vertices in each community is included in the set of keywords and is ranked according to the community qualities. The quality of each community is assessed according to its attributes, such as size and

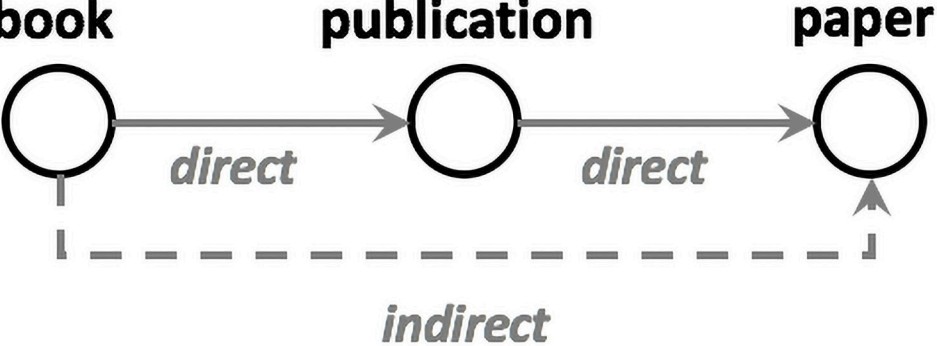

**Fig 1. Illustration of the synonym relationship types.**

diameter. The set of extracted keywords is used to assess topic diversity. Finally, sentiment analysis is conducted to identify the general orientation of the opinions in the text. The proposed approach achieved significant results compared to other keyword extraction techniques. Our primary contributions are as follows. (1) We propose a graph-based text representation approach using word direct and indirect synonym relationships. (2) We extract a representative set of keywords from the text graph based on its structure. We primarily use vertex centrality and the community structure. (3) We analyze the topic diversity and sentiment of the text using the set of extracted keywords.

Topic diversity refers to the presence of multiple, possibly contradictory, topics in a given text [23]. Several diversity dimensions have been discussed in the literature, including diversity in topic, diversity in viewpoint, and diversity in language. Here, we focus on topic diversity, which is the diverse representation of information (by including different ideas, dimensions, beliefs, perspectives, or feelings) on a specific topic. The greater the number of topics in a conversation the more diverse it is. Text sentiment analysis attempts to extract the semantic orientation conveyed in the text, which can be positive, negative, or neutral [23]. Topic diversity and sentiment analysis have many applications in health care, public opinion analysis, social relationship analysis, marketing, and sales predictions [24]. Various methods for topic modeling and extraction [25–27] and sentiment analysis [28, 29] have been discussed in the literature. In this study, we use the structure of the text graph and the set of extracted keywords to assess the topic diversity of a given text and analyze its sentiment.

There are several limitations of the current study. First, the proposed method associates between word pairs based on their synonym relationships and does not consider word contexts. Moreover, identifying the words that actively contribute to the meaning of the text during prerocessing is challenging because the part of speech (POS) taggers are usually trained on a different dataset. Finally, the community detection approach that produces the most accurate keyword set requires further exploration.

## Preliminaries

A graph is a mathematical representation that allows the effective exploration of the relationships between the elements of a system. A given text $T$ can be represented as a directed weighted graph $G = (V, E)$, where $V$ is the set of vertices (words) and $E \subseteq V^2$ is the set of edges. Edges exist between node pairs based on a specific text relation between them.

Each node $u$ has an in-degree $degree_{in}(u)$ that represents the number of edges pointing toward $u$. Moreover, each vertex $u$ has an out-degree $degree_{out}(u)$ that represents the number of edges pointing out from $u$ toward other vertices. The distance between a pair of vertices $u$ and $v$ in $G$ is defined as the number of edges on a shortest path between $u$ and $v$. The weighted distance between $u$ and $v$ is defined as the sum of the weights of all the edges that exist on a shortest path between $u$ and $v$. Here, a shortest path is a path that minimizes the distance between a vertex pair. The *diameter* of a graph $diam(G)$ is the length of a longest shortest path between any two vertices $u$ and $v$ in $G$, i.e., $diam(G) = \max_{u,v \in V}\{d(u, v)\}$.

A *subgraph* $G_W = (W, E_W)$, where $W \subseteq V$ and $E_W = \{uv \in E: u, v \in W\}$, is called the subgraph of $G$ induced by vertex set $W$. A *strongly connected component* in a directed graph is a subgraph in which each vertex is reachable from every other vertex in the subgraph. A *weakly connected component* in a directed graph is a subgraph in which each vertex is reachable from every other vertex in the subgraph despite the direction of the edges. A *singleton vertex $u$* is a vertex with no connections to any other vertex in the graph, i.e., $degree_{in}(u) = degree_{out}(u) = 0$.

In graph theory, centrality measures rank vertices based on their importance in the graph. *Degree centrality* considers the central vertices of the graph as the set of vertices that have the

highest number of connections. *Betweenness centrality* expresses how much effect each vertex has in the communication process between other vertices. Finally, *closeness centrality* considers the graph center as the subset of vertices with the minimum total distance to all other vertices.

The clustering coefficient of a given graph $G$, denoted by $CC(G)$, measures the extent to which vertices tend to cluster together. $CC(G) = \frac{1}{n}\sum_{u=1}^{|V|} CC(u)$, where $CC(u)$ is the clustering coefficient of a single vertex $u$. $CC(u)$ is computed as the proportion of edges among $u$'s neighbors to the number of all possible edges that could exist within the neighborhood of vertex $u$.

A graph *community* refers to a set of vertices that is densely connected internally and loosely connected to other vertices outside the community. *Graph modularity* [30], denoted by $M \in [-1, 1]$, is a graph property that measures the quality of a proposed division of a graph into distinct communities. $M$ is positive when the number of edges between the vertices within the community is high compared to what we would be expected by chance (indicating a better community division) and negative if the number of edges is less than what we would be expected by chance.

Several methods have been proposed for community detection in networks. The *Louvain algorithm* is a greedy algorithm that attempts to optimize the modularity of a network partition. First, the algorithm looks for small communities by optimizing modularity locally. Second, it builds a new network by aggregating vertices within each community. The steps are repeated until a maximum of modularity is attained. This process naturally produces a hierarchical decomposition of the network.

The *Leiden community detection algorithm* [31], which is an extension of the Louvain algorithm, partitions the vertices into different communities that are guaranteed to be connected. The proposed communities are then refined by splitting them further into multiple partitions or merging vertices with a randomly chosen community.

In this work, we use both the Louvain and the Leiden community detection algorithms because they do not require a priori knowledge of the number of communities that will be detected. Moreover, both algorithms have the advantage of finding high quality communities in a time-efficient manner.

## Related work

This work proposes a keyword extraction technique by exploiting the structure of the text graph and the synonym relationships between words. The set of extracted keywords is used to assess the topic diversity and sentiment of the text. In this section, we review relevant keyword extraction, sentiment analysis, and topic diversity approaches with the focus on graph-based text analysis techniques.

### Modeling text as graphs

Modeling text as graphs attempts to uncover text patterns and analyze linguistic properties hidden in the text. Graph-based approaches require transforming the text into a structured format (graph) by identifying the graph vertices and edges. First, subset of the words in the text are selected as vertex candidates. Second, the relationships connecting vertex pairs need to be identified. Relationships between vertex pairs vary from very simple ones such as word co-occurrences (words that appear together, in the same sequence, or within a specific window) [3, 13, 15, 16, 18] to more complex ones such as word semantic relationships [11, 17] and word syntax relationships [14].

Text graph representation has been used for keyword extraction [3, 11–19], text summarization [32], and language classification [33]. Modeling text as graphs has also been used for

text semantic analysis including information retrieval [34, 35] and authorship attribution analysis [36, 37], and word sense disambiguation [38, 39].

Text graph representation can be enhanced using the concept of word embedding [40] in which words are represented by vectors capturing their semantical and contextual features. Vector similarity measures are used to capture the similarity between the words. For example, in a word co-occurrence graph, identifying words that are semantically similar may not be straightforward. For example, "hard" and "difficult" may be mapped into two distinct vertices. In this case, word embedding can be used to map words conveying the same meaning into the same vertex by adding virtual edges between words with similar vectors. Some word embedding strategies include Word2Vec and FastText. The Word2Vec [41] defines dense vector representations of words using a three layer neural network with a single hidden layer. The FastText [42] represents each word as a bag of character n-grams. Therefore, the neural network trains several n-grams of each word. The word vector is the sum of vectors obtained for the character n-grams of the word.

An approach that uses word embedding for keyword extraction was proposed in [43]. The evaluation showed that using word embedding for keyword extraction outperforms many baseline algorithms. Keyword extraction using word embedding was also explored in [44]. First, a word embedding model that integrates local context information of the word graph is used to represent each word. Second, a novel PageRank-based model that incorporates the embedded information is used to rank the candidate words. In [45], the authors investigated whether adding virtual edges using word embedding in co-occurrence graphs may improve the quality of text classification tasks. Their results showed that using word embedding increased the classification performance compared to using traditional co-occurrence graphs.

## Keyword extraction

Keyword extraction refers to the process of detecting the most relevant terms and expressions from a given text. Here, the goal is to summarize the text content and highlight its main topics. Automatic keyword extraction is a key step for multiple text mining applications including summarization, classification, clustering, and topic detection [2, 3]. Keyword extraction techniques range from simple statistical approaches, such as word frequency [5] and word collection and co-occurrence [6], to more advanced machine learning approaches, such as Naive Bayes [8] and support vector machine [9].

Recent keyword extraction methods use both statistics and context information. For example, YAKE [7] relies on the word position and frequency as well as new statistical metrics that capture context information. YAKE calculates five features for each term, i.e., case, position, frequency, relatedness to context, and how often a candidate word appears in different sentences. Then, all features are used to compute a score for each term.

Multiple graph-based text representations have been used for keyword extraction such as word co-occurrence [3, 13, 15, 16, 18], word semantic relationships [11, 17], and syntax relationships [14]. The underlying principle in graph-based keyword extraction is measuring and identifying the most important vertices (words) based on information obtained from the structure of the graph, e.g., vertex centrality [3, 12, 13, 15, 16, 18] and $k$-degeneracy [18, 21].

A comparison of five centrality measures (degree, closeness, betweenness, eigenvector, and TextRank) showed that the simple degree centrality measures achieved results comparable to those of the widely used TextRank [12] algorithm. In addition, it has been shown that closeness centrality outperforms the other centrality measures on short documents [46].

TextRank [12] is a popular graph-based keyword and sentence extraction technique. TextRank uses word co-occurrence relocation to control the distance between word occurrences.

In other words, two lexical units (vertices) are connected if they occur within a window of maximum $N$ words.

Syntactic filters can be used to select lexical units of a certain part of speech to be added to the graph. Vertices are then ranked based on the PageRank algorithm and the top vertices are returned as keywords. In [47], the authors construct the text graph based on word semantic similarity and then use PageRank centrality to extract keywords. [48] introduced a text analysis and visualization method using graph analysis tools to identify the pathways for meaning circulation.

Keyword extraction from a collection of texts using semantic relationship graphs has been discussed in [17]. A graph is first constructed using word co-occurrences and related senses. That is, word relationships are not established only based on a simple co-occurrence, but also as a result of a significant number of occurrences within a document that represents a semantic unit. Then, the relations between words in the obtained graph are enriched with information from WordNet.

Word semantic relations have also been exploited in [49]. The proposed method identifies exemplar words by leveraging clustering techniques to guarantee that the document is semantically covered. First, words are grouped into clusters according to their semantic distances. Then, each cluster is represented by a centroid word. The exemplar words are used to extract the keywords from the document.

Motivated by the fact that both documents and words can be represented by several semantic topics, [50] proposed a keyword extraction technique using multiple random walks specific to various topics. Accordingly, they assigned multiple importance scores to each word. Then, keywords are extracted based on their relevance to the document and their topic coverage.

In [51], the authors proposed a method to extract the main topics in conversations among Twitter users. They created their text graph based on the logical proximity of the concepts. That is, two words become adjacent if they are shared by users directly or through some specific separation degree. Then they use the $k$-core and modularity to isolate the different topics in the text. Measuring the level of bias in a discourse using text graphs is discussed in [52]. The proposed tool creates the text graph based on word co-occurrences. The most influential words and the different topics are identified using betweenness centrality and community detection techniques.

The above graph-based keyword extraction methods suffer from several problems. First, they require the number of keywords as a preset parameter as they are not able to find an optimal number of keywords based on the content of the text. Second, the constructed text graphs rely only mostly on co-occurrence relations ignoring any semantic relationships between the terms in the text.

**Keyword extraction using word synonym relationships.** [53] proposed a keyword extraction algorithm using PageRank on synonym graphs. First, the text is represented as a weighted synonym co-occurrence graph. Then, the PageRank algorithm is used to rank each synonym group. Finally, several top-ranked synonym groups are selected as keywords. Using word synonym relationships for keyword extraction has also been discussed in the literature under the notion of lexical chains [54]. Lexical chains describe sets of semantically related words. Lexical chains can be created using three steps: (1) select a set of candidate words, (2) determine a suitable chain by calculating the semantic relatedness among members of the chain, and (3) if a chain exists, add the word and update the chain; else, create a new chain to fit the word [54, 55]. The second step can be performed using an existing database of synsets, such as the one included in the WordNet corpus [56]. Lexical chains and graph centrality measures were also used for keyword extraction in [55, 57].

## Topic diversity and sentiment analysis

Topic diversity refers to the diverse representation of information on a specific topic by including different ideas, beliefs, perspectives, or feelings. Topic diversity is related to sentiment analysis, opinion mining, and text summarization. Various approaches can be applied to analyze topic diversity. In [58], the authors proposed a natural language processing technique to discover opinion diversity in a text using domain-specific vocabulary. They used two initial lists of positive and negative adjectives. Each list is then expanded using word synonym and antonym relationships. [59] used a graph-based template approach for topic variation detection. Here, the text is represented as semantic subgraphs and best matching subgraphs are used as a template to compare the text in an unsupervised manner.

Topic diversity can also be investigated through the analysis of community structure in graphs. In [4], text is first modeled as a graph of semantic relationships between terms. Then, community detection techniques are used for keyword extraction. The results showed that the terms related to the main topics of the text tend to form several cohesive communities. [60] identified a collection of communities related to a range of topics in Twitter conversation graphs. In [15], noun phrases that represent the main topics are extracted, clustered into topics, and used as vertices in a complete graph. Topics are scored using TextRank [12], and key phrases are extracted by selecting the most representative candidate from each of the top-ranked topics.

Sentiment analysis aims to classify a text as positive, negative, or neutral. Its primary focus is identifying opinion words in the text. Supervised learning techniques with three classes (positive, negative, and neutral) have been used by training research data [29]. For example, customer ratings can be directly translated into a class. Here, a 4–5 star review is considered positive, a 1–2 star review is considered negative, and a review of 3 is considered neutral. A previous study [28] identified opinion sentences in customer reviews about a specific product feature to determine if the review is positive or negative. The authors identified opinion adjectives and determine their sentiment using WordNet. Note that WordNet and other similar sources do not include sentiment information for each adjective word; thus, the authors used the WordNet synonym and antonym sets to predict the sentiment information.

The current paper proposes an unsupervised parameterless domain independent keyword extraction approach using text graph representation. The proposed approach does not require a training dataset labeled by humans. Moreover, because the proposed approach relies on grouping semantically similar candidate words, it can extract a set of keywords that covers the main topics discussed in the text.

## Proposed method

Given a text comprising a collection of text items, such as comments to a tweet or a news article, the proposed method aims to identify a set of keywords in the text, assess the diversity of the text, and analyze its sentiment (the code is included as Supplementary S1 Code). Here, the main idea is to represent the text as a synonym graph and then analyze the graph structure. The proposed method is described in the following steps (Fig 2 describes the proposed method workflow).

### Step 1. Data preprocessing

Generally when people write in a conversational style, their texts tend to be very noisy for any text mining technique. Therefore, the proposed method starts with preprocessing the text based on the following steps.

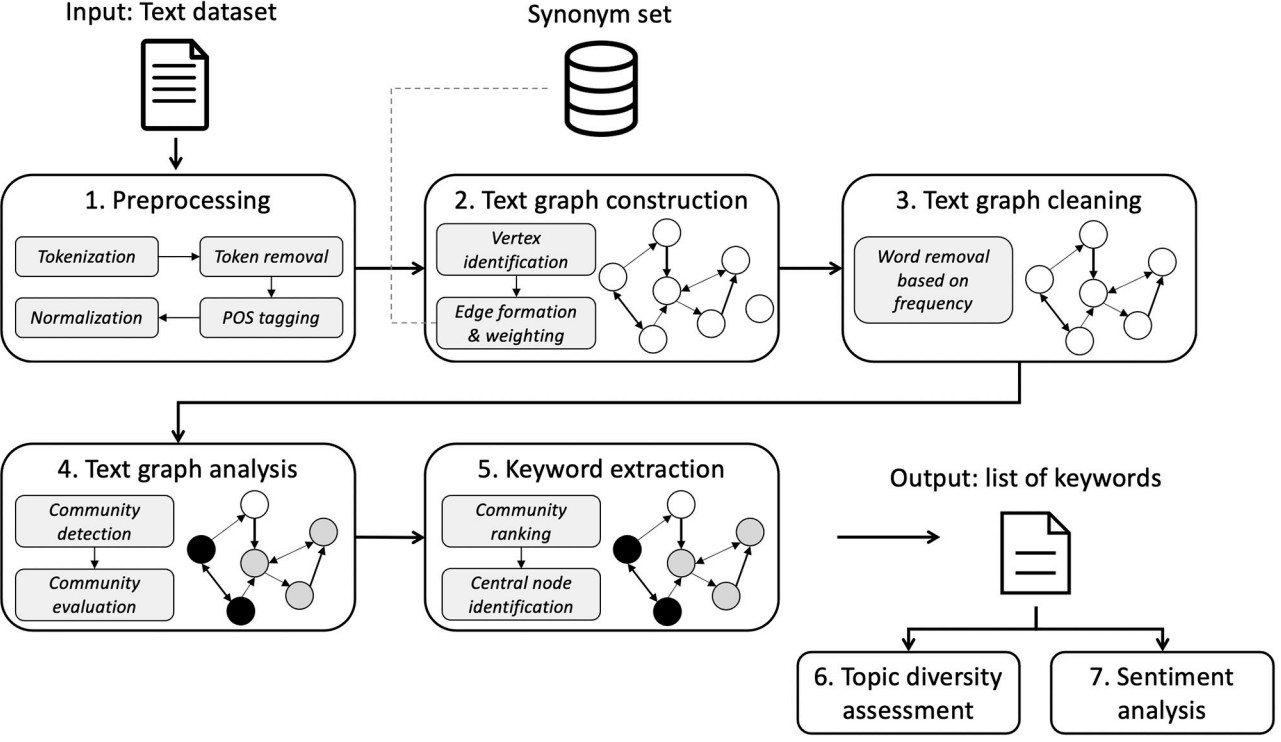

**Fig 2. Proposed method main steps and workflow.**

- Tokenization: Each word/item in the text is treated as a token. A given text $T$ is represented as $T = \{t_1, t_2, \ldots t_i\}$, where $i$ is the number of tokens.

- Token removal: All stop word tokens, non-alphabetic word tokens, and non-English word tokens are removed from the text. Moreover, when the text contains user names (Twitter text, for example), they are removed from the text. In this work, we use the NLTK corpus stopwords [56].

- POS analysis: The part of speech (POS) of each of the remaining tokens is analyzed and only tokens that actively contribute to the meaning of the text are included. Here we include nouns, proper nouns, adjectives, and adverbs (including comparative adjectives and adverbs). In this work, we use the Semantic/syntactic Extraction using a Neural graph Architecture (SENNA) part of speech tagger [61].

- Token normalization: All words are normalized by converting them to their lemmas, i.e., their meaningful base forms. Here we use the WordNet lemmatizer [56].

## Step 2. Text graph construction

The text is converted to a directed weighted graph $G = (V, E)$, where the set of vertices $V$ represents the set of tokens and the set of edges $E$ represents the synonym relationships between word pairs as follows. A directed edge is added between two vertices $u$ and $v$ if vertex (word) $v$ is a synonym of vertex (word) $u$. Edge direction is used to capture the asymmetric relationships between word synonyms [62]. For example, according to the Collins English Dictionary [63], the word "unit" is a synonym of the word "department", but the opposite is not true.

Here, each edge has a weight that represents the strength of the relationship between the pair of words. We assume two relationship types (strengths): direct and indirect.

A word $v$ is a direct synonym of another word $u$ if it has a similar meaning. Word $v$ is an indirect synonym of word $u$ if a word $w$ which is a synonym of $v$ is also a synonym of word $u$. For example, in Fig 1, the word "publication" is a direct synonym of the word "book," while the word "paper" is an indirect synonym of the word "book". Direct synonym relationships between word pairs represent stronger relationships among word pairs compared to indirect synonym relationships. In other words, direct synonym relationships result in higher edge weights.

In this text graph, the in-degree of a vertex $u$ represents how many words have $u$ as their synonym. The out-degree of vertex $u$ represents the number of other words in the text that are synonyms of $u$. In addition, each word $u$ has the *frequency* attribute (denoted $freq(u)$), which represents the number of occurrences of the word.

Given two words $u$ and $v$ and their synonym sets denoted by $syn(u)$ and $syn(v)$, respectively, the edges and their weights in the text graph are assigned as follows.

$$w(e_{uv}) = \begin{cases} 1, \text{if } v \in syn(u). \\ 0.5, \text{if } v \in syn(syn(z_i)), \text{ where } z_i \in syn(u). \\ \infty, \text{otherwise.} \end{cases} \tag{1}$$

where $e_{uv}$ is a directed edge pointing toward vertex $v$, and $w(e_{uv})$ denotes the edge weight. The weights are selected to represent direct and indirect synonym relationships (words that are direct synonyms have a stronger relationship in the text graph). Throughout this work, we use the Networkx Python library (https://networkx.org) to construct and analyze our text graphs.

## Step 3. Text graph cleaning

Unimportant vertices (words) are removed from the text graph. We measure vertex importance based on its degree (number of connections) and its frequency in the text (number of times it occurs within the text). Both measures indicate word important within a given text [64].

Let $S$ be the set of singleton vertices in $G$, where $S = \{u \in V: degree_{in}(u) > 0 \text{ and } degree_{out}(u)\} > 0$. Singleton vertices (vertices with in-degree and out-degree equal to zero) with frequency of one are considered to have a little contribution to the topic; therefore, those vertices are removed from the text graph. However, singleton vertices with frequency greater than one are considered high contributors. This set of singleton vertices forms the set $\bar{S}$, where $\bar{S} = \{u \in S : freq(u) > 1\}$.

## Step 4. Text graph analysis—Community extraction and evaluation

Our goal is to use the structure of the text graph to identify its keywords, which will be used subsequently to assess the text topic diversity and analyze its sentiment. To do so, we partition the vertices of the text graph into distinct communities each of which includes (to an extent) a set of interrelated concepts. We then apply the concept of vertex centrality to extract keywords from each community.

The text graph is first partitioned into communities using one of the community detection algorithms such as the Louvain algorithm [65] and the Leiden algorithm [31], and then the qualities of each community are assessed. We define the following attributes for each community $C_i$ with $|V_i| = n_i$ vertices and $|E_i| = m_i$ edges:

- *size*($C_i$): The community size defined as $size(C_i) = n_i + m_i$. Larger community sizes indicate the existence of more words and relationships within the community, i.e., the same concept (or concepts) was introduced in multiple different ways in the text or most of the concepts are synonyms. Accordingly, the larger the size of a community, the more important it is in the text.

- *weight*($C_i$): The community weight is computed as the sum of the frequencies of all vertices in the community, i.e., $\sum_{j=1}^{n_i} freq(u_j)$. The larger the weight of the community, the more relevant it is in the text.

- *density*($C_i$): The community density is defined as $density(C_i) = \frac{m_i}{n_i(n_i-1)}$, where $0 \leq density$ $(C_i) \leq 1$. Density reports the difference between the number of existing edges and the maximum possible number of edges. Higher densities imply stronger vertex relationships.

- *diam*($C_i$): The community diameter is the length of a longest shortest path between any two vertices in the subgraph induced by the community vertices. Larger community diameters indicate the existence of words that are less relevant to each other.

- *CC*($C_i$): The community clustering coefficient, where $0 \leq CC(C_i) \leq 1$. The clustering coefficient measures the extent to which vertices tend to cluster together. A larger clustering coefficient indicates a community that includes strongly related words. This indicates a stronger synonym relationships between the words.

Communities are sorted lexicographically according to their weights, sizes, and densities. Each community is then assigned a Quality value, which can be "High" or "Low". The Quality of a community $C_i$ is considered High if it achieves at least one of the following: $density(C_i) \geq \delta$ or $CC(C_i) \geq \delta$, where $\delta > 0$ is a threshold for the community quality. Otherwise, the community quality is considered Low.

Community qualities can be enhanced by partitioning its vertices into smaller communities. This process can be applied to communities with lower qualities and repeated iteratively until no further community partition enhancement is possible.

## Step 5. Keywords identification

To extract the set of keywords from the given text, we use the concepts of vertex centrality and community quality. First, the graph communities are ranked based on their attributes: community weight, size, clustering coefficient, and diameter. Then, the set of most central vertices in each community is identified.

Multiple centrality measures can be used to rank vertices. Here, we use the in-degree centrality. Vertices with higher in-degrees are more important since they are synonyms to more words in the text. The most important vertices with respect to the in-degree measure is denoted $in-degree_k^{top}(C_i)$, where $k \geq 1$ is the selected number of top words to be returned from this set. In addition, we compute the sets of medium important vertices and least important vertices for each community (denoted by $in-degree_k^{med}(C_i)$ and $in-degree_k^{least}(C_i)$, respectively).

The set of key words *kw* will be formed by combining the most important vertices in each community. In addition, all important singleton vertices (set $\bar{S}$ in Section 1) will be added. That is:

$$kw = \bigcup_i in-degree_k^{top}(C_i) \cup \bar{S} \tag{2}$$

To increase the precision and the quality of set *kw*, the sets of medium and least important vertices (sets $in - degree_k^{med}(C_i)$ and $in - degree_k^{least}(C_i)$) will be included for communities with low quality. This is important to ensure the completeness of the content because communities with low quality may include words that are not strongly related.

## Step 6. Topic diversity assessment

Topic diversity is assessed using the number of weakly connected components in the text graph $|W|$ and the graph modularity $M$. The number of weakly connected components ($|W|$) shows how strongly related the vertices in the graph are. The graph modularity $M$ indicates how well defined the communities in the graph are. Using the two values, topic diversity in the text graph is assessed as follows.

- When $\frac{|W|}{|V|} \approx 1$, topic diversity is very high since the text contains many topics that are weakly related.

- When $\frac{|W|}{|V|} \ll 1$ and $M \geq 0.65$, topic diversity is high since the text contains multiple topics that are weakly related.

- When $\frac{|W|}{|V|} \ll 1$ and $M < 0.65$, topic diversity is low since the text contains few topics that are closely related.

    The modularity threshold (0.65) was chosen based on previous studies [52, 65].

## Step 7. Sentiment analysis

The overall text sentiment is assessed by identifying the general orientation (polarity) of the set of keywords *kw*. Here, the sentiment of each keyword is assessed using the VADER package [66] and accordingly classified as positive, negative, or neutral. Given a concept *w*, VADER assigns it a polarity score *polarity(w)* that shows its orientation and the orientation level ($-1 \leq$ *polarity(w)*$\leq 1$). The overall text sentiment is assessed in two ways as follows.

1. Determine the cardinalities of the positive, negative, and neutral concepts in the set *kw*. This provides a sentiment analysis overview about the text.

2. Compute the text weighted polarity *P* as

$$P = \sum_{i}^{|kw|} polarity(w_i), \tag{3}$$

    where *kw* is the set of keywords and *polarity(w_i)* is the polarity score of keyword $w_i$ as assigned by VADER.

## An Illustrative example

We use the set of replies to a Tweet posted by CNN on Dec. 29, 2019 (available at https://twitter.com/CNN/status/1210348818492997633) to demonstrate the proposed method. At the time of this analysis, the initial CNN tweet had 87 replies (listed in Supplementary S1 Text). The text was first tokenized into a list of words and preprocessed.

During preprocessing, we used the Semantic/syntactic Extraction using a Neural graph Architecture (SENNA) part of speech tagger [61] to keep only nouns, proper nouns, adjectives, and adverbs. The remaining tokens were then normalized by converting them to their lemmas.

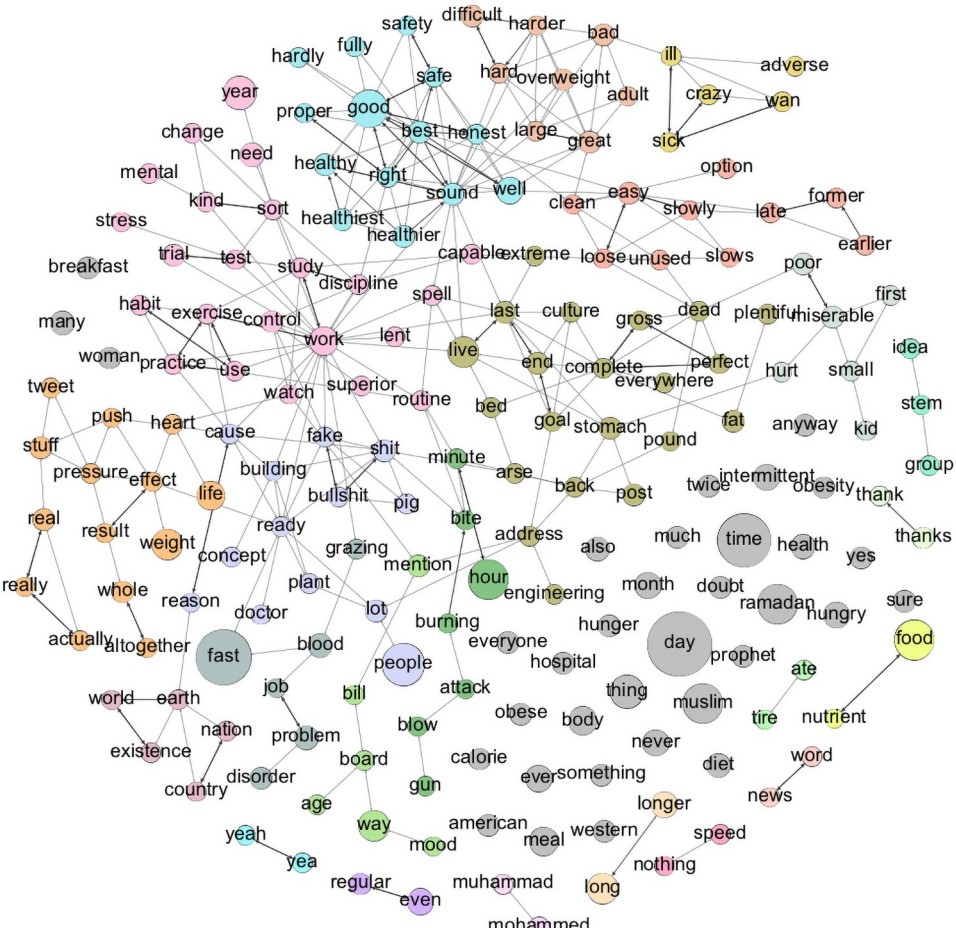

**Fig 3. Text graph for the tweet replies text.** This graph has 193 vertices (34 vertices are singletons), 572 edges, and 45 connected components. 112 edges represent direct synonym relationships, and 460 represent indirect synonym relationships. The size of each vertex reflects its frequency in the original text. Thick and thin edges indicate direct and indirect synonym relationships, respectively.

For example, "I hope the money is going to charity." will become ["hope", "money", "go", "charity"].

Then the associated text graph was constructed. We used the word synonyms from Word-Net corpus [56] using the NLTK toolkit to identify synonym relationships between vertices in the text graph. The unimportant vertices (singletons with frequencies less than two) were removed.

Fig 3 shows the text graph associated with the tweet replies. The text graph has 193 vertices (34 vertices are singletons), 572 edges, and 45 connected components. 112 edges represent direct synonym relationships, and 460 represent indirect synonym relationships. For example, in Fig 3, the word "proper" is a direct synonym of the word "right" and an indirect synonym of the word "best." Table 1 lists the set of important singleton vertices $\bar{S}$.

The qualities of the 22 main communities (partitioned using the Louvain algorithm) in the text graph are summarized in Table 2. We evaluate the quality of each community using its attributes: weight, size, density, diameter, and clustering coefficient. The last column in Table 2 shows the quality of each community based on its attributes. Quality is classified as "High" or "Low". The quality of a community $C_i$ is considered high if it achieves at least one of

**Table 1. High contributor singleton vertices in the text graph and their frequencies.**

| vertex $u$ | $freq(u)$ | vertex $u$ | $freq(u)$ | vertex $u$ | $freq(u)$ | vertex $u$ | $freq(u)$ |
|---|---|---|---|---|---|---|---|
| ramadan | 10 | muslim | 10 | month | 4 | day | 21 |
| time | 16 | many | 3 | meal | 5 | intermittent | 4 |
| also | 3 | twice | 2 | health | 3 | breakfast | 2 |
| never | 4 | hungry | 4 | doubt | 2 | western | 2 |
| something | 2 | prophet | 2 | yes | 2 | sure | 2 |
| woman | 2 | much | 2 | hospital | 2 | obesity | 2 |
| anyway | 2 | calorie | 2 | everyone | 2 | american | 2 |
| body | 5 | ever | 4 | thing | 7 | obese | 2 |
| hunger | 2 | diet | 3 | | | | |

**Table 2. Summary of the qualities of the 22 main communities in the text graph in Fig 3.**

| $C_i$ | weight ($C_i$) | $\|V_{C_i}\|$ | $\|E_{C_i}\|$ | size ($C_i$) | density ($C_i$) | diam ($C_i$) | CC ($C_i$) | Most central vertices (in degree) | Medium central vertices (in degree) | Least central vertices (in degree) | Quality |
|---|---|---|---|---|---|---|---|---|---|---|---|
| | 33 | 23 | 48 | 71 | 0.09 | 9 | 0.3 | study, sort, discipline | practice, habit, change | bill, age, lent | Low |
| | 31 | 21 | 72 | 93 | 0.17 | 5 | 0.3 | complete, last, end | address, perfect, bed | extreme, plentiful, engineering | Low |
| | 30 | 14 | 86 | 100 | 0.5 | 2 | 0.86 | good, best, sound | honest, well, healthiest | safety, fully, hardly | High |
| | 26 | 5 | 8 | 13 | 0.4 | 4 | 0 | blood, job, problem | | disorder, fast | Low |
| | 25 | 13 | 42 | 55 | 0.27 | 4 | 0.4 | ready, shit, building | cause, bullshit, lot | concept, doctor, reason | Low |
| | 25 | 13 | 32 | 45 | 0.21 | 6 | 0.36 | pressure, effect, stuff | tweet, really, heart | life, result, whole | Low |
| | 16 | 7 | 12 | 19 | 0.29 | 6 | 0 | bite, minute, blow | attack | burning, hour, gun | Low |
| | 12 | 8 | 38 | 46 | 0.68 | 3 | 0.83 | harder, large, great, | hard, bad, | overweight, adult, difficult | High |
| | 11 | 2 | 2 | 4 | 1 | 1 | 0 | food, nutrient | | | High |
| | 10 | 2 | 2 | 4 | 1 | 1 | 0 | long, longer | | | High |
| | 9 | 8 | 26 | 34 | 0.46 | 3 | 0.63 | loose, easy, slowly | clean, slows | late, unused, option | High |
| | 8 | 6 | 12 | 18 | 0.4 | 3 | 0.25 | miserable, small, first | | hurt, kid, poor | Low |
| | 6 | 2 | 2 | 4 | 1 | 1 | 0 | regular, even | | | High |
| | 5 | 5 | 14 | 19 | 0.7 | 2 | 0.7 | ill, crazy, wan | | sick, adverse | High |
| | 5 | 5 | 12 | 17 | 0.6 | 2 | 0.87 | earth, existence, world | | nation, country | High |
| | 3 | 3 | 4 | 7 | 0.67 | 2 | 0 | stem, group, idea | | | Low |
| | 3 | 2 | 2 | 4 | 1 | 1 | 1 | thank, thanks | | | High |
| | 2 | 2 | 2 | 4 | 1 | 1 | 1 | speed, nothing | | | High |
| | 2 | 2 | 2 | 4 | 1 | 1 | 1 | yea, yeah | | | High |
| | 2 | 2 | 2 | 4 | 1 | 1 | 1 | ate, tire | | | High |
| | 2 | 2 | 2 | 4 | 1 | 1 | 1 | word, news | | | High |
| | 2 | 2 | 2 | 4 | 1 | 1 | 1 | muhammed, mohammed | | | High |

**Table 3. Synonym relationships and clustering coefficients (*CC*) of four communities in the text graph.** The numbers represent relationship strengths (a strength of 1 is assigned between direct synonyms and 0.5 is assigned between indirect synonyms).

(**a**) *CC* = 0

|  | blood | job | problem | grazing | fast | disorder |
|---|---|---|---|---|---|---|
| blood |  | 0.5 |  | 0.5 | 0.5 |  |
| job | 0.5 |  | 1 |  |  |  |
| problem |  | 1 |  |  |  | 0.5 |
| grazing | 0.5 |  |  |  |  |  |
| fast | 0.5 |  |  |  |  |  |
| disorder |  |  | 0.5 |  |  |  |

(**b**) *CC* = 0.25

|  | miserable | small | first | hurt | kid | poor |
|---|---|---|---|---|---|---|
| miserable |  | 0.5 | 0.5 | 0.5 |  | 1 |
| small | 0.5 |  | 0.5 |  | 0.5 |  |
| first | 0.5 | 0.5 |  |  |  |  |
| hurt | 0.5 |  |  |  |  |  |
| kid |  | 0.5 |  |  |  |  |
| poor | 1 |  |  |  |  |  |

(**c**) *CC* = 0.7

|  | ill | sick | wan | adverse | crazy |
|---|---|---|---|---|---|
| ill |  | 1 | 0.5 | 0.5 | 0.5 |
| sick | 1 |  | 1 |  | 1 |
| wan | 0.5 | 1 |  |  | 0.5 |
| adverse | 0.5 |  |  |  |  |
| crazy | 0.5 | 1 | 0.5 |  |  |

(**d**) *CC* = 0.87

|  | nation | country | earth | world | existence |
|---|---|---|---|---|---|
| nation |  | 1 | 0.5 |  |  |
| country | 1 |  | 0.5 |  |  |
| earth | 0.5 | 0.5 |  | 1 | 0.5 |
| world |  |  | 1 |  | 1 |
| existence |  |  | 0.5 | 1 |  |

the following: *density*($C_i$)≥0.5 or *CC*($C_i$)≥0.5. Otherwise, the community quality is considered low.

First, communities are sorted lexiclavically according to their weights and sizes. Then each community is assigned a quality rank according to its density, clustering coefficient, and diameter.

Table 3(a)–3(d) show the synonym relationships within four communities. The community presented in Table 3(b) has a low clustering coefficient compared to the community in Table 3(c). Higher clustering coefficients indicate that the majority of words in the community are connected by either direct or indirect synonym relationships and vise verse. The concepts in Table 3(c) are more related to one another; as shown in Fig 3, they form almost a star graph. The concepts within the community in Table 3(b) also form a star but with weaker ties between the vertices (smaller weights). This indicates that it contains words that are not closely related such as the words "miserable" and "poor". Similarly, the density of a community

reflects the relationship strength among the words in the community. Note that the clustering coefficient of the community in Table 3(c) is 0.7, while the clustering coefficient of the community in Table 3(b) is 0.25. Another community attribute that can be used to evaluate quality is the diameter. Generally, the length of the diameter (with respect to the number of vertices in the community) correlates negatively with its quality. In other words, shorter diameters indicate stronger communities and vice versa. For example, community ▉ with 23 vertices has a diameter of 3, while community ▉ with 100 vertices has a diameter of 2.

The set of keywords *kw* for *k* = 1 and all singleton vertices for our example is

*kw* = {study, practice, bill, complete, address, extreme, good, blood, disorder, ready, cause, concept, pressure, tweet, life, bite, attack, burning, harder, food, long, loose, miserable, hurt, regular, ill, earth, stem, thank, speed, yea, ate, word, muhammed, ramadan, muslim, month, day, body, time, many, meal, intermittent, ever, also, twice, health, breakfast, thing, never, hungry, doubt, western, obese, something, prophet, yes, sure, hunger, woman, much, hospital, obesity, diet, anyway, calorie, everyone, american}.

Community extraction using the Leiden algorithm for this example is briefly discussed in S1 Appendix. The set of keywords using iterative community extraction is shown in S2 Appendix.

The number of weakly connected components in the text graph $|W|$ = 45, the graph modularity $M$ = 0.75, and the number of vertices $|V|$ = 193. That is, $\frac{|W|}{|V|} \ll 1$ and $M \geq 0.65$, which suggests high topic diversity within the text.

We also analyzed the sentiment of the text. In our example, the set of keywords includes 68 words. The sentiment analysis shows that 8 words are positive, 10 are negative, and 50 are neutral. The weighted polarity $P$ = −0.24 suggests that the text is slightly negative in orientation.

## Evaluation

To assess the keyword extraction performance of the proposed technique, we compared its performance to that of two keyword extraction techniques: TextRank [12] (a graph-based technique) and YAKE [7] (a statistical-based technique) using several different datasets. TextRank uses word co-occurrences to control the distance between word occurrences in creating the text graph. Then it uses eigenvector centrality to rank each term. YAKE computes a score for each term based on five features: case, position, frequency, relatedness to context, and how often a candidate word appears in different sentences.

Three performance measures were used as key concept extraction evaluation metrics: precision (*Pr*), recall (*Re*), and *F-score* defined as follows.

$$Pr = \frac{|\{Relevant\} \cap \{Retrieved\}|}{|\{Retrieved\}|} \quad (4)$$

$$Re = \frac{|\{Inter\_Relevant\} \cap \{Retrieved\}|}{|\{Inter\_Relevant\}|} \quad (5)$$

$$F-measure = 2 \times \frac{Pr \times Re}{(Pr + Re)} \quad (6)$$

We also used a metric similar to the Pyramid [67] for our evaluation. The Pyramid method [67] creates a pyramid using the human annotated keywords. The set of keywords extracted by each method is then compared to the pyramid. Each keyword *w* is assigned a pyramid score *ps* based on the number of human annotators who selected it. The higher the pyramid score, the

higher the keyword is in the pyramid. A system's oracle score *os*, where $0 \leq os \leq 1$, is computed by adding the pyramid scores of the keywords generated by the system.

In our performance analysis, we let the number of keywords returned by each method contribute to a method's oracle score. We define the *weighted-score* as follows.

$$weighted-score = \frac{os}{\sum_{w=1}^{N} ps(w)} * \frac{1}{|U| + 0^{|U|}},$$

where *U* = {*Retrieved*} − {*Keywords*} and *N* is the number of unique keywords extracted by the human extractors.

Two evaluation tasks were performed: evaluation using human extractors and evaluation using publicly available annotated datasets. For each dataset, we used the community partition (Louvain or Leiden) that yielded the best result.

## Evaluation using human extractors

In this evaluation task, three human extractors were asked to extract an unspecified number of keywords from a given text dataset. The human extractors were instructed to extract keywords based on importance and relevance to the topic. Then, the intersection of the three keyword sets was determined.

Table 4 shows three sets of keywords extracted by three human extractors for the text dataset that includes the Tweet replies for the Tweet in the Illustrative example. The performances of the proposed method, TextRank, and YAKE are shown in Table 5. In Table 5, *Relevant* represents the set of keywords that appear in at least one of the lists extracted by the human extractors. *Inter_Relevant* represents the set of keywords that appears in the intersection set of the human extractor lists. Precision represents the probability that a key concept is relevant

**Table 4. Keywords identified by three human extractors.**

| | |
|---|---|
| Extractor 1 (*E1*) | fast, ramadan, muslim, month, health, eat, body, islam, meal, starving, intermittent, food, money, mood, control, discipline, piety, tired, prophet, muhammed, breakfast, mental, physical, masculinity, water, irresponsible, hungry, bush, anorexia, starve, years, pressure, sugar, drop, bad, fall, obese, safe, nonsense, diabetes, burning, calorie, hours, difficult, stupid, unhealthy, affect, good, hunger, study, weight, american, news |
| Extractor 2 (*E2*) | fast, ramadan, eat, healthier, stop, develop, lose, weight, calorie, save, muslim, right, live, refreshing, water, drink, body, food, hunger, obesity, prophet, muhammed, suger, meal, medicine, diet, stress, skin, month, islam, solve, metabolism, soul, starving, fat, stomach, intermittent, works, day, study, control, extended, health, cured, allergies, compulsory, optional, teaching, western, discipline, piety, breakfast, mental, physical, guidelines |
| Extractor 3 (*E3*) | fast, ramadan, muslim, month, healthier, weight, body, soul, several, science, proving, islam, eat, intermittent, lose, works, food, control, keto, obese, inhabitants, holy, prophet, muhammed, day, year, crazy, moderation, doubt, women, starve, good, long, time, hungry, live, fake, news, miserable, blood, pressure, sugar, yoga, obesity, anorexia, nutrients, upset, diet, stress, calorie, fat, abstaining, american, sure, tiered, bad |
| *E1∩E2∩E3* | fast, ramadan, muslim, month, eat, weight, body, islam, calorie, intermittent, food, control, prophet, muhammed |
| Proposed method | **study**, practice, bill, complete, address, extreme, **good**, **blood**, disorder, ready, cause, concept, **pressure**, tweet, life, bite, attack, **burning**, harder, <u>**food**</u>, **long**, loose, **miserable**, hurt, regular, ill, earth, stem, speed, yea, **ate**, word, <u>**muhammed**</u>, <u>**ramadan**</u>, <u>**muslim**</u>, <u>**month**</u>, **day**, <u>**body**</u>, **time**, many, **meal**, <u>**intermittent**</u>, ever, also, **health**, **breakfast**, thing, never, **hungry**, **doubt**, **western**, **obese**, something, <u>**prophet**</u>, yes, **sure, hunger, woman, much, hospital, obesity, diet, anyway**, <u>**calorie**</u>, **everyone, american** |

Relevant words (keywords that appear in at least one of the lists extracted by the human extractors) appear in bold.

Inter_Relevant words (keywords that appear in the intersection set of the human extractor lists) are underlined.

**Table 5. Performance comparison of the proposed method with TextRank and YAKE.**

| Method | N | Precision (*Pr*) | Recall (*Re*) | *F-score* |
|---|---|---|---|---|
| TextRank | 34 | 44.1% | 35.7% | 39.5% |
| YAKE | 65 | 36.9% | 35.7% | 35.2% |
| YAKE* | 51 | 39.2% | 35.7% | 36.5% |
| Proposed method | 68 | 45.6% | 64.3% | 53.4% |
| Proposed method* | 51 | 43.1% | 50.0% | 46.3% |

*N* represents the number of keywords a method returned. Proposed method* represents the set of keywords after removing singleton vertices with frequencies ≤2.

given that it is returned by a system. Recall represents the probability that a relevant key concept is returned [68, 69].

Table 5 shows the performance of two versions of the proposed method; first, when all keywords are included (set *kw* as discussed in Section 4.4.2), and second, set *kw* after removing singleton vertices with frequencies ≤2. For comparison, two YAKE versions are also listed in Table 5: YAKE, which represents all keywords returned by this technique, and YAKE*, which represents the 51 most important concepts.

Moreover, this experiment was conducted with eight more text datasets: five Tweet replies datasets, two CNN news datasets, and one speech dataset. The Tweet replies datasets are: TRAVEL-BAN, AIRPODS, GENE-TECH, JEFF-BEZOS, and INT-FASTING. All five datasets were collected from Twitter. Each dataset contains the set of Tweet replies posted under each Tweet. The numbers of replies associated with each tweet are 69, 72, 30, 161, and 87. The CNN news datasets are: BIDEN and COVID-19. The last dataset (KING-SPEECH) is a transcript of Martin Luther King Jr's "I have dream" speech. Table 6 lists the main statistical data of each dataset.

For each dataset, we used the keyword set extracted by the human extractors to compare the performance of the proposed method against TextRank and YAKE (see Table 7). Table 7 shows two sets of comparisons. The first comparison considers all extracted keywords by each approach. The second comparison considers only the first $\bar{N}$ keywords, where $\bar{N}$ is the number of unique keywords extracted by all human extractors.

In Table 7, precision represents the number of correctly matched words to the total number of extracted words. That is, $Pr = \frac{\text{True\_positives}}{\text{True\_positives} + \text{False\_positives}}$. Recall represents the

**Table 6. Statistical and text graph data of each dataset.** Number of words and number of tokens denote the number of words in the dataset before and after preprocessing respectively. Direct edges and indirect edges represent the number of direct and indirect synonym relationships between words in the text graph respectively.

| Dataset | Dataset data | | Text graph data | | | | | |
|---|---|---|---|---|---|---|---|---|
| | No. of words | No. of tokens | \|V\| | \|E\| | Direct edges | Indirect edges | No. of singletons | No. of conn components |
| TRAVEL-BAN | 812 | 418 | 113 | 172 | 31 | 141 | 20 | 26 |
| AIRPODS | 980 | 498 | 123 | 143 | 37 | 106 | 23 | 34 |
| GENE-TECH | 593 | 300 | 88 | 90 | 18 | 72 | 13 | 23 |
| JEFF-BEZOS | 2197 | 1083 | 255 | 521 | 105 | 416 | 45 | 52 |
| INT-FASTING | 1499 | 762 | 193 | 572 | 112 | 460 | 34 | 45 |
| BIDEN | 1532 | 791 | 230 | 510 | 91 | 139 | 38 | 47 |
| COVID-19 | 1875 | 1031 | 211 | 338 | 55 | 283 | 49 | 58 |
| KING-SPEECH | 1657 | 808 | 210 | 266 | 55 | 211 | 45 | 58 |

**Table 7. Comparison of multiple keyword extraction methods.**

| Dataset | Method | N | Pr | Re | F | N | Pr | Re | F | weighted-score |
|---|---|---|---|---|---|---|---|---|---|---|
| TRAVEL-BAN | TextRank | 180 | 51.1 | 100 | 67.7 | 147 | 50 | 100 | 66.7 | 0.007 |
| | YAKE | 274 | 48.3 | 100 | 65.2 | 147 | 59.2 | 100 | 74.4 | 0.011 |
| | Proposed method ($k = 1$) | 54 | 51.9 | 40 | 45.2 | 54 | 51.9 | 40 | 45.2 | 0.007 |
| | Proposed method ($k = 2$) | 81 | 39.5 | 40 | 39.8 | 81 | 39.5 | 40 | 39.8 | 0.005 |
| | Proposed method ($k = 3$) | 91 | 39.6 | 40 | 39.8 | 91 | 39.6 | 40 | 39.8 | 0.005 |
| AIRPODS | TextRank | 164 | 51.3 | 60 | 55.3 | 146 | 50.8 | 40 | 44.7 | 0.008 |
| | YAKE | 275 | 41.3 | 60 | 48.9 | 146 | 44.8 | 60 | 51.3 | 0.006 |
| | Proposed method ($k = 1$) | 60 | 50.8 | 40 | 44.8 | 60 | 50.8 | 40 | 44.8 | 0.009 |
| | Proposed method ($k = 2$) | 87 | 44.8 | 40 | 42.3 | 87 | 44.8 | 40 | 42.3 | 0.007 |
| | Proposed method ($k = 3$) | 102 | 43.1 | 40 | 41.5 | 102 | 43.1 | 40 | 41.5 | 0.006 |
| GENE-TECH | TextRank | 180 | 51.1 | 100 | 67.7 | 147 | 50 | 100 | 66.7 | 0.015 |
| | YAKE | 274 | 48.3 | 100 | 65.2 | 147 | 59.2 | 100 | 74.4 | 0.011 |
| | Proposed method ($k = 1$) | 54 | 51.9 | 40 | 45.2 | 54 | 51.9 | 40 | 45.2 | 0.01 |
| | Proposed method ($k = 2$) | 81 | 39.5 | 40 | 39.8 | 81 | 39.5 | 40 | 39.8 | 0.006 |
| | Proposed method ($k = 3$) | 91 | 39.6 | 40 | 39.8 | 91 | 39.6 | 40 | 39.8 | 0.007 |
| JEFF-BEZOS | TextRank | 366 | 46.8 | 40 | 43.1 | 258 | 45.7 | 20 | 27.8 | 0.004 |
| | YAKE | 585 | 39.9 | 100 | 57.1 | 263 | 48.8 | 50 | 49.4 | 0.004 |
| | Proposed method ($k = 1$) | 83 | 45.8 | 20 | 27.8 | 83 | 45.8 | 20 | 27.8 | 0.004 |
| | Proposed method ($k = 2$) | 117 | 41.9 | 20 | 27.1 | 117 | 41.9 | 20 | 27.1 | 0.003 |
| | Proposed method ($k = 3$) | 145 | 41.4 | 20 | 27 | 145 | 41.4 | 20 | 27 | 0.003 |
| INT-FASTING | TextRank | 220 | 23.4 | 63.6 | 34.3 | 107 | 29.5 | 54.5 | 38.3 | 0.005 |
| | YAKE | 409 | 22.9 | 90.9 | 36.5 | 107 | 41.9 | 72.7 | 53.2 | 0.008 |
| | Proposed method ($k = 1$) | 67 | 46.3 | 72.7 | 56.6 | 67 | 46.3 | 72.7 | 56.6 | 0.01 |
| | Proposed method ($k = 2$) | 100 | 34 | 63.6 | 44.3 | 100 | 34 | 63.6 | 44.3 | 0.006 |
| | Proposed method ($k = 3$) | 123 | 29.3 | 63.6 | 40.1 | 107 | 25.2 | 45.5 | 32.5 | 0.004 |
| BIDEN | TextRank | 258 | 72.3 | 60 | 65.6 | 258 | 72.3 | 60 | 65.6 | 0.01 |
| | YAKE | 421 | 59.8 | 60 | 59.9 | 302 | 66.4 | 60 | 63.1 | 0.007 |
| | Proposed method ($k = 1$) | 82 | 64.6 | 50 | 56.4 | 82 | 64.6 | 50 | 56.4 | 0.007 |
| | Proposed method ($k = 2$) | 128 | 62.5 | 50 | 55.6 | 128 | 62.5 | 50 | 55.6 | 0.005 |
| | Proposed method ($k = 3$) | 158 | 56.3 | 60 | 58.1 | 158 | 56.3 | 60 | 58.1 | 0.005 |
| COVID-19 | TextRank | 285 | 75.3 | 69.6 | 72.3 | 285 | 75.3 | 69.6 | 72.3 | 0.01 |
| | YAKE | 455 | 64.1 | 91.3 | 75.3 | 342 | 68.6 | 91.3 | 78.4 | 0.007 |
| | Proposed method ($k = 1$) | 87 | 64.4 | 26.1 | 37.1 | 87 | 64.4 | 26.1 | 37.1 | 0.005 |
| | Proposed method ($k = 2$) | 125 | 57.6 | 28.3 | 37.9 | 125 | 57.6 | 28.3 | 37.9 | 0.005 |
| | Proposed method ($k = 3$) | 159 | 56 | 37 | 44.5 | 159 | 56 | 37 | 44.5 | 0.004 |
| KING-SPEECH | TextRank | 244 | 57.7 | 55.6 | 56.6 | 206 | 57.6 | 55.6 | 56.6 | 0.007 |
| | YAKE | 409 | 45.7 | 66.7 | 54.2 | 206 | 53.9 | 55.6 | 54.7 | 0.006 |
| | Proposed method ($k = 1$) | 91 | 51.6 | 66.7 | 58.2 | 91 | 51.6 | 66.7 | 58.2 | 0.007 |
| | Proposed method ($k = 2$) | 126 | 44.4 | 55.6 | 49.4 | 126 | 44.4 | 55.6 | 49.4 | 0.005 |
| | Proposed method ($k = 3$) | 149 | 40.3 | 66.7 | 50.2 | 149 | 40.3 | 66.7 | 50.2 | 0.004 |

number of correctly matched words to the total number of assigned words. That is, $Re = \frac{\text{True positives}}{\text{True positives} + \text{False negatives}}$. The table shows the performance of the proposed method using three different values of $k$ (the number of words extracted from each community in the text graph).

**Table 8. Statistical and text graph data of each abstract in the HULTH dataset.** Number of words and number of tokens denote the number of words in the dataset before and after preprocessing respectively. Direct edges and indirect edges represent the number of direct and indirect synonym relationships between words in the text graph respectively.

| Dataset | Dataset data | | | Text graph data | | | | | |
|---|---|---|---|---|---|---|---|---|---|
| | No. of words | No. of tokens | No. of keywords | \|V\| | \|E\| | Direct edges | Indirect edges | No. of singletons | No. of conn components |
| ABSTRACT-1 | 308 | 204 | 47 | 44 | 18 | 4 | 14 | 23 | 29 |
| ABSTRACT-2 | 126 | 73 | 29 | 16 | 5 | 1 | 4 | 9 | 11 |
| ABSTRACT-3 | 183 | 114 | 32 | 34 | 34 | 12 | 22 | 8 | 12 |
| ABSTRACT-4 | 200 | 112 | 40 | 27 | 30 | 5 | 25 | 6 | 8 |
| ABSTRACT-5 | 156 | 93 | 55 | 23 | 13 | 4 | 9 | 8 | 12 |

## Evaluation using annotated datasets

As another experiment to evaluate the performances of the proposed method and the other existing methods used for comparison, we used a set of publicly available human annotated datasets from the *Inspec* database [10]. The dataset includes a collection of abstracts and the corresponding manually assigned keywords. The abstracts are from Computer Science and Information Technology journal papers. Two sets of keywords are assigned for each abstract: controlled (restricted to a given thesaurus) and uncontrolled (freely assigned by the annotators). Following [10, 12], we use the uncontrolled set of keywords for our comparisons.

First, we extracted the set of human annotated keywords for five abstract datasets (see Table 8). Then we used the keyword set to compare the performance of the proposed method against the baselines: TextRank, and YAKE (the number of extracted keywords was limited to the number of keywords assigned by human annotators). The results are shown in Table 9.

In Table 9, Precision represents the number of correctly matched words to the total number of extracted words and Recall represents the number of correctly matched words to the total number of assigned words. Table 9 shows the performance of the proposed method using three different values of *k* for the proposed method.

## Discussion

Table 5 shows the results of the proposed method and the two baselines when compared using the keywords extract by three human extractors. The proposed method achieves better results compared to the two baseline methods when all keywords are included. The performance of the proposed method is slightly affected by the removal of singleton vertices with low frequencies ($\leq 2$).

Table 7 shows the results for all text datasets. Overall, the proposed method shows good results. Using *Precision*, the proposed method ourperforms all baselines on TRAVEL-BAN, GENE-TECH, and INT-FASTING. On AIRPODS and JEFF-BEZOS the proposed method provides comparable results to both baselines. On BIDEN, COVID-19, and KING-SPEECH datasets, the proposed method fails to do better than the baselines. This can be justified by the number of keywords extracted by each method. In all datasets, the two baselines extracts a large number of keywords. For example the number of keywords extracted by TextRank is about three times the number of keywords extracted by the proposed method in almost all datasets. Similarly, the number of keywords extracted by YAKE is about four times the number of keywords extracted by the proposed method. Overall, the proposed method extracts a concise set of keywords without requiring the number of keywords as a preset parameter.

Using the *weighted-score*, the proposed method achieves comparable results to both baselines with a fraction of keywords.

**Table 9. Comparison of multiple keyword extraction methods against human annotators.**

| Dataset | Method | N | Pr | Re | F |
|---|---|---|---|---|---|
| ABSTRACT-1 | TextRank | 55 | 28.8 | 40.4 | 33.6 |
| | YAKE | 100 | 24.2 | 32 | 27.5 |
| | Proposed method ($k = 1$) | 34 | 32 | 47.1 | 38.1 |
| | Proposed method ($k = 2$) | 42 | 32.3 | 47.6 | 38.5 |
| | Proposed method ($k = 3$) | 42 | 32.3 | 47.6 | 38.5 |
| ABSTRACT-2 | TextRank | 27 | 31.6 | 46.2 | 37.5 |
| | YAKE | 49 | 35.6 | 55.2 | 43.2 |
| | Proposed method ($k = 1$) | 14 | 41.7 | 71.4 | 52.6 |
| | Proposed method ($k = 2$) | 16 | 40.7 | 68.8 | 51.2 |
| | Proposed method ($k = 3$) | 16 | 40.7 | 68.8 | 51.2 |
| ABSTRACT-3 | TextRank | 56 | 30.4 | 43.8 | 35.9 |
| | YAKE | 82 | 31.9 | 46.9 | 38 |
| | Proposed method ($k = 1$) | 22 | 33.3 | 50 | 40 |
| | Proposed method ($k = 2$) | 31 | 32.6 | 48.4 | 39 |
| | Proposed method ($k = 3$) | 34 | 32 | 46.9 | 38 |
| ABSTRACT-4 | TextRank | 40 | 27.3 | 37.5 | 31.6 |
| | YAKE | 70 | 24.5 | 23.5 | 28 |
| | Proposed method ($k = 1$) | 18 | 25 | 33.3 | 28.6 |
| | Proposed method ($k = 2$) | 25 | 21.9 | 28 | 24.6 |
| | Proposed method ($k = 3$) | 25 | 21.9 | 28 | 24.6 |
| ABSTRACT-5 | TextRank | 33 | 31.3 | 45.5 | 37 |
| | YAKE | 56 | 29.5 | 41.8 | 34.6 |
| | Proposed method ($k = 1$) | 16 | 42.9 | 75 | 54.5 |
| | Proposed method ($k = 2$) | 21 | 40 | 66.7 | 50 |
| | Proposed method ($k = 3$) | 21 | 40 | 66.7 | 50 |

Increasing the number of extracted keywords (by increasing the value of $k$) does not seem to improve the performance of the proposed method. This highlights the importance of synonym relationships between words (communities in the text graph can be represented by a single word).

Table 9 shows results for the TextRank, YAKE, and the proposed method compared against keywords extracted by human annotators. Similar trends as the ones shown in Table 7 can be observed. First, the number of keywords extracted by the proposed method is smaller than the number of keywords returned by TextRank and YAKE. In fact, the number of keywords extracted by the proposed method is smaller than the number of keywords annotated by the human annotators. This is because our method takes into account the semantic relationships between the words. This is crucial when there is a limit on the number of returned keywords. For example, the dataset ABSTRACT-5 includes two pairs of direct synonym words: "velocity" and "speed" and "procedure" and "function".

Second, Table 9 shows that the proposed method outperforms TextRank and YAKE. However, increasing the number of extracted keywords (by increasing $k$) does not seem to improve the performance of the proposed method. Again, this shows that including semantically related words does not increase the accuracy of keyword extraction algorithms.

The highest $F$-score achieved is about 55% and the average is 34.9%. Keyword extraction is a highly subjective task and an $F$-score of 100% is infeasible [70]. For example, in the human annotated abstract datasets, some keywords are not present in the abstracts as human indexers

had access to full-length documents when assigning the keywords [10]. For example, the number of absent keywords from the ABSTRACT-4 and ABSTRACT-5 datasets analyzed above are 10 and 16 respectively. This implies that the highest a method could theoretically achieve is 100% precision for both datasets, 75% recall ABSTRACT-4 and 71% recall for ABSTRACT-5. This gives a maximum *F*-score of 87% for ABSTRACT-4 and 83% for ABSTRACT-5.

A number of limitations of the proposed method must be noted. First, in English (and many other languages), the meaning of a word usually depends on its context. However, the proposed method associates word pairs based on their synonym relationships and does not consider context. For example, the words "regular" and "even" are direct synonyms; thus, they make their own community. The word "regular" appeared twice in the text in the following comments:

- "I went from 285+ to 165 at the age of 50 . . . I fast for 20+ hrs and will never go back to a "regular" eating schedule."

- "Naaah when I am hungry I just tuck in. The yoga stuff is not for regular folk. Good day snowflakes!"

  The word "even" appeared four times in the following comments:

- "Irresponsible post. Need to preface this with excluding women. A lot of women. Don't do well with even a 12 hour fast."

- "I don't even want to live another day let along any longer! Thanks to fake news!"

- "Americans can't even give up guns and you want them to give up food? They're overweight for a reason."

- "who told you you gotta eat three times or even more a day and who told you you gotta eat each time till you filled up complete day after day."

  Considering context in relation to these comments, "regular" and "even" carry two different meanings and should not be in the same community.

The other limitation is related to the words' POS and the POS taggers. The SENNA tagger [61] is used to identify the POS of the words and retain those that carry the core text content. In our case, we selected nouns, proper nouns, adjectives, and adverbs. The POS taggers may not always identify the correct POS due to the differences between the training dataset and the current dataset. Typically, those differences are usually related to the different uses of words in English. For example, identifying the correct POS of a word ending with "ing" can be problematic.

## Conclusion

The goal of keyword extraction is to identify the concepts that describe the main topics discussed in a conversation. Keyword extraction can provide insights about the topics discussed within the text. Keyword extraction approaches can be categorized as statistical, machine learning, linguistic, and graph-based approaches. Graph-based keyword extraction approaches capture more structural information about the text compared to other text analysis techniques.

In this work, we extract the keywords in a given text, assess its topic diversity, and analyze its sentiment using graph representation of the text and the synonym relationships between the words. We first partition the text graph into different communities and then identify the most central vertices as keywords. The quality of each community is assessed according to its attributes, such as the number of vertices and edges, its diameter, and its clustering coefficient. The community quality indicates the strength of relationship between the words in the

community. We first sort the communities according to their qualities and then extract the most central vertices in each community using the degree centrality measure. We also include the set of single vertices with high frequencies to the set of keywords.

The motivation behind our work is to overcome the limitations of other graph-based keyword extraction approaches, primarily their dependence on word co-occurrences only for the text graph construction and their user parameter requirements. Our basic concept can be improved by including collections (words that appear adjacent to one another) or co-occurrences (words that appear together within the text but not necessarily adjacent) analyses.

Word synonym relationships connect words with semantic associations. Here we used two synonym relationship types: direct and indirect. A word is a direct synonym of another word if it has a similar meaning. A word is an indirect synonym of another word if they are both synonyms of a third one. Word synonym relationships are used as edge weights to indicate the strength of relationship between them (direct synonym relationships are stronger to compared to the indirect ones).

The proposed method has a number of limitations. First, the proposed method associates word pairs based on their synonym relationships and does not consider context. However, the meaning of English words usually depends on the context. Second, the part of speech (POS) taggers, which are used to select words that actively contribute to the meaning of the text during prerocessing, may not always identify the correct POS due to the differences between the training dataset and the current dataset. Typically, those differences are related to the different uses of words in English. Finally, the community detection approach that results in the best keyword set is yet to be explored.

As future work, we plan to extend this word relationship formation by including higher degree synonym relationships among words. New studies considering other vertex centrality measures to rank the words in each community can be proposed. Moreover, community detection approaches that allow for overlapping communities need to be considered. Further analysis need to be conducted to identify the best community detection algorithm that can be used with synonym graphs. Finally, using word embedding and virtual edges to improve the performance of the proposed approach need to be investigated.

## Supporting information

**S1 Code. Code used for text graph construction and analysis is available at https://github. com/halrashe/Topic-Diversity.**
(TXT)

**S1 Text. The replies associated with the tweet in the Illustrative example section.** Twitter handles were anonymized.
(PDF)

**S1 Appendix. Community extraction using the Leiden algorithm.**
(PDF)

**S2 Appendix. Keywords using iterative community extraction.**
(PDF)

## Acknowledgments

The author thank Maram Bahareth for her insight during the early stages of this research. The author also thank the Deanship of Scientific Research and RSSU at King Saud University for their technical support.

## Author Contributions

**Conceptualization:** Hend Alrasheed.

**Data curation:** Hend Alrasheed.

**Formal analysis:** Hend Alrasheed.

**Investigation:** Hend Alrasheed.

**Methodology:** Hend Alrasheed.

**Resources:** Hend Alrasheed.

**Software:** Hend Alrasheed.

**Supervision:** Hend Alrasheed.

**Validation:** Hend Alrasheed.

**Visualization:** Hend Alrasheed.

**Writing – original draft:** Hend Alrasheed.

**Writing – review & editing:** Hend Alrasheed.

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
