## [Decision Letter · Decision Letter 0]

4 Mar 2021

PONE-D-21-00554

Word synonym relationships for text analysis: A graph-based approach

PLOS ONE

Dear Dr. Alrasheed,

Thank you for submitting your manuscript to PLOS ONE. After careful consideration, we feel that it has merit but does not fully meet PLOS ONE’s publication criteria as it currently stands. Therefore, we invite you to submit a revised version of the manuscript that addresses the points raised during the review process.

We look forward to receiving your revised manuscript.

Kind regards,

Diego Raphael Amancio

Academic Editor

PLOS ONE

Journal Requirements:

Reviewers' comments:

Reviewer's Responses to Questions

**Comments to the Author**

1. Is the manuscript technically sound, and do the data support the conclusions?

Reviewer #1: Partly

Reviewer #2: Partly

Reviewer #3: Partly

2. Has the statistical analysis been performed appropriately and rigorously? 

Reviewer #1: I Don't Know

Reviewer #2: I Don't Know

Reviewer #3: Yes

3. Have the authors made all data underlying the findings in their manuscript fully available?

Reviewer #1: Yes

Reviewer #2: No

Reviewer #3: Yes

4. Is the manuscript presented in an intelligible fashion and written in standard English?

Reviewer #1: Yes

Reviewer #2: Yes

Reviewer #3: Yes

5. Review Comments to the Author

Reviewer #1: This is a potentially publishable paper, but it needs a major revision. First, in related work, it's not clear why the kinds of work reviewed were selected. I struggled to understand why I was reading about sentiment analysis, for example, when the stated problem was keyword extraction. It became clear later but at the time I read it I was confused.

Second, related work is summarized but but no conclusions are drawn from the summary. By the time we get to the proposed method, it is absolutely unclear if there are any problems with existing methods or what problems the proposed method is trying to solve. It makes the proposed method look like a solution in search of a problem.

Third, a lot of details that should have been discussed in the proposed solution didn't appear until the illustrative example (for example, what software was used for POS tagging). It's a very odd way to organize a paper.

Fourth, the solution is tested on tweets, many of which are ungrammatical (you can see this in examples at the end). POS taggers tend to fail on ungrammatical text. Perhaps this is part of the reason the F1 was not better.

Fifth, in addition, the datasets used are tiny, consisting of at most only 1000 tokens. Why not test against one of the Linguistic Data Consortium's corpora? https://catalog.ldc.upenn.edu/

Sixth, I was quite confused by the decision to use TF/IDF as a way to evaluate the proposed solution. TF/IDF is a method of identifying keywords. If it counts as a standard for evaluation, why do we need the proposed solution?

Finally, the F1s reported for the new method are better than those for the comparison methods, but they are still terrible. Why should we publish a paper on a method that's only 53% effective? I could imagine possible arguments, but I don't see any here.

P.S.: there is a typo in Table 6 "tokenz"

Reviewer #2: I am a social scientist with some experience with networks and text analysis, but little in machine learning.

The paper is carefully and well-written, but there are a few anomalies - the text refers to 49 replies, the appendix includes many more than this.

1) Traag (2019) points to arbitrariness in the Louvain algorithm, and in my experience results are unstable. The Leiden algorithm appears more robust, it should at least be explored here.

2) Why are directed graphs used? Can the authors provide an example of where the synonym graph is directed or asymnmetrical? In Table 3, all are symmetrical. (And the entries in panels c and d are the same).

3) The larger communities are typically lower in 'quality,' I am wondering if an iterative approach might be applied (i.e., if a community is large and low quality, look for communities within it)

4) In Table 4, please add emphasis for the 'inter-relevant terms'

5) The paper addresses keyword-identification as a general problem, but all of the datasets are drawn from Twitter. The lack of generalizability beyond this should be at least acknowledged.

re the availability of data - links to and or the availability of the TRAVEL-BAN, AIRPODS, 442 GENE-TECH, JEFF-BEZOS, and INT-FASTING databases to be made clear

Reviewer #3: 1- This paper describes an interesting work on identifying keywords using graph strcture.

2- This paper can be accepted for publication, but some issues should be addressed before

this it can be accepted.

- Some related works on keyword analysis should also be mentioned.

- It is not necessary to define simple measurements such as precision and recall

- Is your method only valid for tweets. Could you please provide a discussion

on how to apply in other documents?

- Related research on modeling texts as complex networks should be mentioned.

- This includes application in summarization, authorship analysis and other

areas. This could be included e.g. in the related works section. Examples:

networks and texts: doi: 10.1016/J.INS.2018.02.047 and doi: 10.1016/j.physa.2011.12.011

networks and statistical metrics: 10.1016/J.PHYSA.2018.03.013 and doi: 10.1016/j.chaos.2021.110679

- It is not clear the criteria for choosing a specific community detection method.

- It would be interesting to mention that embeddings could improve your method.

Recently approaches using virtual edges to improve the classification text networks

have been proposed with embeddings. This possibility, along with related works on this

subject should be mentioned.

- Could increase or apply your dataset in other domains like Yelp, for example?

6. PLOS authors have the option to publish the peer review history of their article (what does this mean?). If published, this will include your full peer review and any attached files.

Reviewer #1: No

Reviewer #2: No

Reviewer #3: No

---

## [Author Response · Author response to Decision Letter 0]

14 May 2021

Point-by-point response to the reviewers’ comments and concerns.

 Response to Reviewer 1 Comments

We thank the reviewer for their reading of the manuscript and for their constructive comments. We have taken the comments into consideration to improve the quality of the manuscript. Please find below a point by point response to each comment.

Comment 1.1: First, in related work, it's not clear why the kinds of work reviewed were selected. I struggled to understand why I was reading about sentiment analysis, for example, when the stated problem was keyword extraction. It became clear later but at the time I read it I was confused.

Response 1.1: We added a couple of sentences at the beginning of the Related Work Section in the revised manuscript (lines: 127-129) that states the most important elements of the proposed work.

Comment 1.2: Second, related work is summarized but but no conclusions are drawn from the summary. By the time we get to the proposed method, it is absolutely unclear if there are any problems with existing methods or what problems the proposed method is trying to solve. It makes the proposed method look like a solution in search of a problem.

Response 1.2: We changed the manuscript as suggested by summarizing the problems with existing graph-based keyword extraction methods at the end of the Keyword Extraction subsection (lines: 228-232). We also added a summary of the advantages of the proposed method compared to existing approaches at the end of the Related Work Section (lines: 277-281).

Comment 1.3: Third, a lot of details that should have been discussed in the proposed solution didn't appear until the illustrative example (for example, what software was used for POS tagging). It's a very odd way to organize a paper. Think about adding a pipeline

Response 1.3: The names of software tools used in each step were added whenever appropriate. We added the POS tagger name (lines: 302-303), the word lemmatizer name (lines: 305-306), and the source we used to identify stop words in the text (line: 298). We also added the software used for graph construction and analysis (lines: 333-334).

A pipeline showing the proposed method workflow was added (Figure 2 and lines: 287-288 in the revised manuscript).

Comment 1.4: Fourth, the solution is tested on tweets, many of which are ungrammatical (you can see this in examples at the end). POS taggers tend to fail on ungrammatical text. Perhaps this is part of the reason the F1 was not better.

Response 1.4: Other document types were added to the Evaluation Section (lines: 540-547), Tables 6, and 8. We added two news documents and one speech transcript (Tables 6). We also added five abstract datasets (Table 8). The proposed approach was applied on all the newly added datasets. The results are listed in Tables 7 and 9.

Comment 1.5: Fifth, in addition, the datasets used are tiny, consisting of at most only 1000 tokens. Why not test against one of the Linguistic Data Consortium's corpora? https://catalog.ldc.upenn.edu/

Response 1.5: We added larger datasets (BIDEN, COVID-19, and KING-SPEECH) in Table 6 in the revised manuscript. However, all dataset sizes tend to be on the smaller side due to the evaluation process used in this work. That is, because human extractors were invited to extract keywords from each dataset, larger datasets, which can be tedious to work with, were not include.

Comment 1.6: Sixth, I was quite confused by the decision to use TF/IDF as a way to evaluate the proposed solution. TF/IDF is a method of identifying keywords. If it counts as a standard for evaluation, why do we need the proposed solution?

Response 1.6: We changed the Evaluation Section completely (lines: 519-520). We removed the “evaluation against TF-IDF” part. Instead, we invited human extractors to identify keywords in each dataset. Then we evaluated the performance of the proposed method and the other existing methods against the sets of keywords extracted by the human readers (Subsection: Evaluation using human extractors and Table 7 - lines: 548-552).

Moreover, we added another evaluation task: evaluation using a publicly available human annotated dataset that included a set of scientific paper abstracts and their assigned keyword sets (Subsection: Evaluation using annotated datasets and Table 9 - lines: 560-575). Discussing the new evaluation results was added to the Discussion Section in the revised manuscript (lines: 582- 621).

Comment 1.7: Finally, the F1s reported for the new method are better than those for the comparison methods, but they are still terrible. Why should we publish a paper on a method that's only 53% effective? I could imagine possible arguments, but I don't see any here.

Response 1.7: A discussion on the possible reason of low F-scores has been included in the Discussion Section (lines: 613-621).

Comment 1.8: P.S.: there is a typo in Table 6 "tokenz"

Response 1.8: The typo has been fixed in Table 6 in the revised manuscript.

Response to Reviewer 2 Comments

We thank the reviewer for their reading of the manuscript and for their constructive comments. We have taken the comments into consideration to improve the quality of the manuscript. Please find below a point by point response to each comment.

Comment 2.1: The paper is carefully and well-written, but there are a few anomalies - the text refers to 49 replies, the appendix includes many more than this.

Response 2.1: The typo was fixed in the revised manuscript (line: 434). Comment 2.2: Traag (2019) points to arbitrariness in the Louvain algorithm, and in my experience

results are unstable. The Leiden algorithm appears more robust, it should at least be explored here.

Response 2.2: We changed the manuscript as suggested by adding the definition of the Leiden community detection algorithm to the Preliminaries Section (lines: 118-121). We also used the Leiden algorithm as well as the Louvain algorithm to identify communities in the text graph throughout our work (lines: 354-355, 487-488, and 520-521).

The vertices of the text graph in the Illustrative example were partitioned using both algorithms (lines: 452-453 and the Appendix in the revised manuscript).

Comment 2.3: Why are directed graphs used? Can the authors provide an example of where the synonym graph is directed or asymnmetrical? In Table 3, all are symmetrical. (And the entries in panels c and d are the same).

Response 2.3: We added a brief discussion and an example that shows the importance of directions in synonym graphs (lines: 311-314 in Step2: Text graph construction).

Moreover, we fixed the entries of Table 3 (d) in the revised manuscript. Now all entries reflect the relationships between the words as shown in Figure 3.

Comment 2.4: The larger communities are typically lower in 'quality,' I am wondering if an iterative approach might be applied (i.e., if a community is large and low quality, look for communities within it)

Response 2.4: We added a brief discussion on the possibility of enhancing the qualities of the obtained communities to Step 4 (Text graph analysis – Community extraction and evaluation) (lines: 383-385). We also show the set of keywords when the iterative keyword extraction is used in Tables 11 and 12 (Appendix) (lines: 488-489 in the revised manuscript).

Comment 2.5: In Table 4, please add emphasis for the 'inter-relevant terms'

Response 2.5: The manuscript was changed as suggested by adding emphasis for the ‘inter- relevant’ terms in Table 4. All inter_relevant terms are underlined in Table 4 in the revised manuscript.

Comment 2.6: The paper addresses keyword-identification as a general problem, but all of the datasets are drawn from Twitter. The lack of generalizability beyond this should be at least acknowledged.

Response 2.6: Other document types were added to the Evaluation Section (lines: 540-547), Tables 6, and 8. We added two news documents and one speech transcript (Tables 6). We also added five abstract datasets (Table 8). The proposed approach was applied on all the newly added datasets. The results are listed in Tables 7 and 9.

Comment 2.7: The availability of data - links to and or the availability of the TRAVEL-BAN, AIRPODS, 442 GENE-TECH, JEFF-BEZOS, and INT-FASTING databases to be made clear

Response 2.7: All text datasets used in this work were uploaded to the manuscript’s GitHub repository. We added a remark about this in the Supplementary information in the revised manuscript (lines 688-689).

Response to Reviewer 3 Comments

We thank the reviewer for their reading of the manuscript and for their constructive comments. We have taken the comments into consideration to improve the quality of the manuscript. Please find below a point by point response to each comment.

Comment 3.1: Some related works on keyword analysis should also be mentioned.

Response 3.1: More related work on keyword analysis has been added to the Related Work Section

of the manuscript (lines: 204-219). References 17, 49, and 50 in the revised manuscript.

Comment 3.2: It is not necessary to define simple measurements such as precision and recall. Response 3.2: A new measurement to compare the performances of the proposed approach and the baselines (named the weighted-score metric) was added. The new metric is defined in the Evaluation Section (lines: 508-518) and used in Table 7 (last column) and in the Discussion Section (lines: 594-595).

Comment 3.3: Is your method only valid for tweets. Could you please provide a discussion on how to apply in other documents?

Response 3.3: Other document types were added to the Evaluation Section (lines: 540-547), Tables 6 and 8. In Table 6, we added two news documents and one speech transcript. In Table 8, we added five abstract datasets. The proposed approach was applied on all added datasets. The results are listed in Tables 7 and 9.

Comment 3.4: Related research on modeling texts as complex networks should be mentioned. This includes application in summarization, authorship analysis and other areas. This could be included e.g. in the related works section. Examples: networks and texts: doi: 10.1016/J.INS.2018.02.047 and doi: 10.1016/j.physa.2011.12.011 networks and statistical metrics: 10.1016/J.PHYSA.2018.03.013 and doi: 10.1016/j.chaos.2021.110679.

Response 3.4: A new subsection named “Modeling text as graphs” was added to the Related Work Section of the revised manuscript (lines: 132-168). In this subsection, we discuss using graphs to represent texts and present the most common text graph representations. We also list the suggested applications that exploited text graph representations along with appropriate references (the suggested references were very helpful. They have been added to this part of the manuscript).

Comment 3.5: It is not clear the criteria for choosing a specific community detection method.

Response 3.5: A brief justification of the choice of the community detection algorithms was added to the Preliminaries Section (lines: 122-125). We also added a remark about the importance of investigating more community detection algorithms to the Conclusion (lines: 680-683).

Comment 3.6: It would be interesting to mention that embeddings could improve your method. Recently approaches using virtual edges to improve the classification text networks have been proposed with embeddings. This possibility, along with related works on this subject should be mentioned.

Response 3.6: We added a brief discussion on using word embedding and virtual edges to the Related Work Section (lines: 146-168). We added a couple of references that used word embedding for keyword extraction and a reference that used word embedding for virtual edges addition (references: 43, 44, 45 in the revised manuscript).

Also, we added a remark about the possibility of improving the results of the proposed approach using word embedding and virtual edges to the Conclusion (lines: 682-683).

Point 3.7: Could increase or apply your dataset in other domains like Yelp, for example?

Response 3.7: Five text datasets from Computer Science and Information Technology journal papers were added to the Evaluation Section (lines: 560-575), Tables 8 and 9 in the revised manuscript.

---

## [Decision Letter · Decision Letter 1]

5 Jul 2021

PONE-D-21-00554R1

Word synonym relationships for text analysis: A graph-based approach

PLOS ONE

Dear Dr. Alrasheed,

Thank you for submitting your manuscript to PLOS ONE. After careful consideration, we feel that it has merit but does not fully meet PLOS ONE’s publication criteria as it currently stands. Therefore, we invite you to submit a revised version of the manuscript that addresses the points raised during the review process.

We look forward to receiving your revised manuscript.

Kind regards,

Diego Raphael Amancio

Academic Editor

PLOS ONE

Journal Requirements:

Additional Editor Comments (if provided):

It would be interesting to add to relevant sections of text (e.g. introduction, conclusion) the limitations of the method so the reader can be aware

of potential problems and advantages of using other approaches such as embedding techniques.

Reviewers' comments:

Reviewer's Responses to Questions

**Comments to the Author**

1. If the authors have adequately addressed your comments raised in a previous round of review and you feel that this manuscript is now acceptable for publication, you may indicate that here to bypass the “Comments to the Author” section, enter your conflict of interest statement in the “Confidential to Editor” section, and submit your "Accept" recommendation.

Reviewer #2: All comments have been addressed

2. Is the manuscript technically sound, and do the data support the conclusions?

Reviewer #2: Yes

3. Has the statistical analysis been performed appropriately and rigorously? 

Reviewer #2: Yes

4. Have the authors made all data underlying the findings in their manuscript fully available?

Reviewer #2: Yes

5. Is the manuscript presented in an intelligible fashion and written in standard English?

Reviewer #2: Yes

6. Review Comments to the Author

Reviewer #2: I appreciate the authors' thoroughness in addressing the concerns raised in the prior version. If they continue this line of inquiry, I urge them to consider other approaches of community extraction (e.g., k-cliques) that allow for overlapping communities, as this could conceivably lead to a more naturalistic set of keywords. But that is for the next paper, not this one.

7. PLOS authors have the option to publish the peer review history of their article (what does this mean?). If published, this will include your full peer review and any attached files.

Reviewer #2: No

---

## [Author Response · Author response to Decision Letter 1]

7 Jul 2021

Comment 1.1: It would be interesting to add to relevant sections of text (e.g. introduction,

conclusion) the limitations of the method so the reader can be aware of potential problems and

advantages of using other approaches such as embedding techniques.

Response 1.1: A brief discussion on the limitation of the proposed method was added to the

Introduction Section (lines: 72-77) and to the Conclusion Section (lines: 686-694).

Comment 2.1: I appreciate the authors' thoroughness in addressing the concerns raised in the prior

version. If they continue this line of inquiry, I urge them to consider other approaches of

community extraction (e.g., k-cliques) that allow for overlapping communities, as this could

conceivably lead to a more naturalistic set of keywords. But that is for the next paper, not this one.

Response 2.1: A note about this was added to the Conclusion Section in the revised manuscript

(lines: 697 and 698) as a possible future work for research in this direction.

Comment 3.1: Please review your reference list to ensure that it is complete and correct. If you

have cited papers that have been retracted, please include the rationale for doing so in the

manuscript text, or remove these references and replace them with relevant current references. Any

changes to the reference list should be mentioned in the rebuttal letter that accompanies your

revised manuscript. If you need to cite a retracted article, indicate the article’s retracted status in

the References list and also include a citation and full reference for the retraction notice.

Response 3.1: The reference list entries have been reviewed. The format of the majority of

references have been updated to match the reference style of the journal. Moreover, we have added

the doi and link to references whenever possible in the References Section of the updated

manuscript.

---

## [Editor Report · Decision Letter 2]

12 Jul 2021

Word synonym relationships for text analysis: A graph-based approach

PONE-D-21-00554R2

Dear Dr. Alrasheed,

We’re pleased to inform you that your manuscript has been judged scientifically suitable for publication and will be formally accepted for publication once it meets all outstanding technical requirements.

Kind regards,

Diego Raphael Amancio

Academic Editor

PLOS ONE
---

## [Editor Report · Acceptance letter]

16 Jul 2021

PONE-D-21-00554R2 

Word synonym relationships for text analysis: A graph-based approach 

Dear Dr. Alrasheed:

I'm pleased to inform you that your manuscript has been deemed suitable for publication in PLOS ONE. Congratulations! Your manuscript is now with our production department. 

Kind regards, 

on behalf of

Dr. Diego Raphael Amancio 

Academic Editor

PLOS ONE